# Biomechanically Tunable Nano-Silica/P-HEMA Structural Hydrogels for Bone Scaffolding

**DOI:** 10.3390/bioengineering8040045

**Published:** 2021-04-04

**Authors:** Raffaella Aversa, Relly Victoria Petrescu, Florian Ion T. Petrescu, Valeria Perrotta, Davide Apicella, Antonio Apicella

**Affiliations:** 1Advanced Materials Lab, Department of Architecture and Industrial Design, Second University of Naples, Abazia di San Lorenzo, 81031 Aversa, Italy; raffaella.aversa@unicampania.it (R.A.); valeria.perrotta@unicampania.it (V.P.); 2IFToMM, ARoTMM, Bucharest Polytechnic University, Splaiul Independenței 313, 060042 Bucharest, Romania; fitpetrescu@gmail.com (R.V.P.); rvvpetrescu@gmail.com (F.I.T.P.); 3CalabroDental, Via Enrico Fermi 5C, 88900 Crotone, Italy; dott.davide.apicella@outlook.it

**Keywords:** biomimetic hydrogels, hybrid nanocomposites, anomalous sorption, tissue engineering

## Abstract

Innovative tissue engineering biomimetic hydrogels based on hydrophilic polymers have been investigated for their physical and mechanical properties. 5% to 25% by volume loading PHEMA-nanosilica glassy hybrid samples were equilibrated at 37 °C in aqueous physiological isotonic and hypotonic saline solutions (0.15 and 0.05 M NaCl) simulating two limiting possible compositions of physiological extracellular fluids. The glassy and hydrated hybrid materials were characterized by both dynamo-mechanical properties and equilibrium absorptions in the two physiological-like aqueous solutions. The mechanical and morphological modifications occurring in the samples have been described. The 5% volume nanosilica loading hybrid nanocomposite composition showed mechanical characteristics in the dry and hydrated states that were comparable to those of cortical bone and articular cartilage, respectively, and then chosen for further sorption kinetics characterization. Sorption and swelling kinetics were monitored up to equilibrium. Changes in water activities and osmotic pressures in the water-hybrid systems equilibrated at the two limiting solute molarities of the physiological solutions have been related to the observed anomalous sorption modes using the Flory-Huggins interaction parameter approach. The bulk modulus of the dry and glassy PHEMA-5% nanosilica hybrid at 37 °C has been observed to be comparable with the values of the osmotic pressures generated from the sorption of isotonic and hypotonic solutions. The anomalous sorption modes and swelling rates are coherent with the difference between osmotic swelling pressures and hybrid glassy nano-composite bulk modulus: the lower the differences the higher the swelling rate and equilibrium solution uptakes. Bone tissue engineering benefits of the use of tuneable biomimetic scaffold biomaterials that can be “designed” to act as biocompatible and biomechanically active hybrid interfaces are discussed.

## 1. Introduction

The last few years have experienced significant advancements in hydrogel preparation in the direction of combining various organic and inorganic components to improve hydrogel physicochemical properties, enabling them to be cutting-edge biomaterials to translate into clinical applications [1,2].

In particular, several reviews have underlined the increasing development of new bone scaffolding materials with improved biocompatibility, non-immunogenicity and toxicity, with controlled degradable absorption and degradation rate that best match the formation of new bone. Such materials [1,2] have been described to present adequate porosities combined with structural properties to form three-dimensional structures with a large surface volume ratio able to host osteogenic cells and growth factors. The enrichment with bioactive molecules, such as extracellular matrix proteins, adhesive peptides, growth factors, hormones, improved physicochemical, mechanical and biological properties of polymeric scaffolds.

Recent advances on hydrogels as hosting systems for cells and growth factors for bone tissue engineering have been extensively reviewed for their preparation, characterization, and application/evaluation [2]. In order to make hydrogel more suitable for the local treatment of bone diseases, polymeric hydrogel preparations in combination with synthetic materials have excellent biocompatibility and biomechanical properties. 

It is then necessary to establish a complete method to evaluate the hydrogel’s properties and biocompatibility with the human body.

The Authors developed novel hybrid structural materials for application in bone tissue engineering presenting anomalous sorption behaviours characterized by a transition from hard glassy to rubber state in the presence of aqueous environments that could be potentially used as bone scaffolding materials. Namely, hybrid nanocomposites based on fumed amorphous silica nanoparticles embedded in a hydrophilic poly-(hydroxyl-ethyl-methacrylate) (pHEMA) were deeply investigated [1,2,3,4,5].

The addition of amorphous silica improved the self-organization of the hydrophilic polymeric network promoting hydrogen bonding of the polymeric chains with the hydrophilic nanoparticles. The resulting initially glassy nanocomposites undergo significant hydration and swelling when exposed to aqueous environments, resulting in more rigid and transparent hydrogel with surprisingly improved mechanical strength [5] that overcomes one of the major disadvantages of the use of hydrogels in applications where structural mechanical properties are needed. Such more resilient hybrid hydrogels have been shown to possess biomimetic and osteoconductive properties that are mandatory in the design of mechanically bioactive bone scaffolds [4,5].

Bone remodelling processes repair the damage removing and replacing the damaged tissues with new bone; this healing process can be strongly favoured by using biocompatible and biomechanically active hydrogels that can be “designed” to reproduce bone compatible and biomimetic structural properties.

In healthy conditions, bone growth and remodelling cooperate to define the more efficient functional structure. Natural biomechanics and bone growth dynamic equilibrium is completely altered when a material stiffer than bone is implanted (such is the case of metal implants) [6].

Loads on bones cause bone strains that generate signals that some osteoblast cells can detect and respond to. Threshold ranges of such signals are involved in the control of modelling and remodelling [7,8,9].

Remodelling processes repair the injury by removing and replacing the damaged tissues with new bone. Moreover, overloading (or under-loading) alters such phenomenon [9]. Early studies by Wolff (1892) stated that mechanics could determine changes in the architecture of bones [10]. Frost mathematically expressed these reactions of the bone tissue to given stimuli to quantitatively assess bone deformations postulated by Wolff [11]. Remodelling processes repair the damage removing and replacing the damaged tissues with new bone. Different studies proved that the micro-damage threshold is about 3000 micro epsilon strains [6,8,11].

The structural hydrogels have been proposed as mechanically bioactive interfaces for implants’ early osteointegration. However, the correct utilization of any synthetic or natural hydrogel to be used in biomedical applications should require complete investigation and characterization of mechanical and physiological fluids’ equilibrium sorption characteristics.

The hydrogel interactions with aqueous environments can affect structural as well as chemo-physical materials properties. Due to the different origin, function and clinical pathologies, in fact, physiological water may present different compositions and properties that need to be investigated in order to understand any possible hydrogel material properties modification.

Generally, the diffusion of small molecules in polymers above their glass transition temperature or in glassy polymers with a low chemical affinity with the diffusing species could by described by ordinary Fick’s law. This first limiting transport process, which can be mathematically described for a planar geometry by an initially linear relationship between the weight gain and the square root of time (linearity is maintained up to 50% of the equilibrium maximum uptake) and by a smooth and continuous concentration profile through the sheet thickness [12].

Conversely, more complex anomalous transport behaviours can be observed when physical phenomena due to the strong interactions between molecules and hosting polymer interactions provides the rate determining transport modes. These complex sorption behaviours include time-dependent boundary conditions, penetrant molecules, concentration-dependent diffusion coefficients, osmotic stresses generation in the penetrated polymer and solvent-induced polymer physical relaxation [13,14,15,16,17].

In such cases, swelling and significant changes in the transport mechanism could be observed as a consequence of the depression of the glass transition of the increasingly plasticized absorbing polymer below the test temperature. Namely, a second limiting transport process (named Case II, since it is other than that exclusively driven by diffusion) is described to occur during the sorption of the high affinity molecules in glassy polymers [15,18].

In Case II transport mode, the weight gain of a planar sample immersed in a liquid or vapor is linear in time until equilibrium is reached [18]. Moreover, a discrete discontinuity between the almost uniform penetrant concentration in the outer swollen layers and the unpenetrated central glassy core. However, thicker materials, such as those utilized in bone scaffolding, may show a more complex behaviour driven by the different contributions to the absorbing process, polymer relaxation and diffusive phenomena [14,16]. 

The initially dry and glassy hydrophilic PHEMA-nanosilica hybrid composites developed by our group have been described to show a similar complex swelling response in the presence of the physiological fluids when used in in vivo animal testing [19,20]. The adaptive properties of bone can benefit the use of scaffold biomimetic materials that adequately respond to the physiological fluids properties changes [19,20,21,22,23]. The basic phenomena describing the biomechanically active hybrid response to physiological chemical modification are discussed in the present paper.

## 2. Materials and Methods

### 2.1. Materials

A commercial 2-hydroxyethyl methacrylate (HEMA), obtained from Sigma-Aldrich Chemicals Co., (St. Louis, MO, USA) has been used as hydrophilic matrix. Fumed silicon dioxide (Aerosil 300 Degussa, Germany) with a mean diameter of 7 nm and specific surface area of 300 m^2^·g^−1^ was utilized as a bioactive filler. The α-α’ azo-iso-butyric-nitrile (AIBN), obtained from Fluka (Milan, Italy) has been utilized as an initiator of the radical polymerization reaction. HEMA monomers were mixed with the fumed silica in the ratio of 5%, 10%, 15%, 20% and 25% by volume.

Fine dispersion of nanosilica has been obtained by further high-speed mixing the solution with a homogenizer. When a transparent homogeneous liquid was obtained, the free radical initiator (AIBN) was added at a concentration of 0.1% *w*/*w* based on the weight of the monomer mixture.

After the dissolution of AIBN, the solution was poured between two glass plates covered with transparent films (3M Visual Systems Products, Europe, France) and lined with a silicon rubber frame (thickness of 2.5 mm) to obtain a uniform pHEMA slabs.

The resin was degassed before transferring in the 2.5 mm thick planar molds and then polymerized in a temperature-controlled oven set at 60 °C for 24 h and then post-cured for 1 h at 90 °C.

### 2.2. Mechanical Characterization

The elastic shear tests, measured on dry and hydrate states at different concentrations of NaCl solutions of the PHEMA-nanosilica hybrids were performed using a dynamic mechanical shear analyzer (DMA-Mettler-Toledo, Zurich, Switzerland). The complex shear moduli were measured at a constant frequency of 10Hz and in isothermal conditions of 37 °C. The as prepared glassy samples with different nanosilica loadings were further desiccated under a vacuum at a temperature of 60 °C for 24 h prior to mechanical testing. In the shear test mode, 10 mm diameter discs and two samples cut from the polymer 2.5 mm thickness plate were placed between three steel discs forming a symmetrical sandwich geometry working under controlled oscillatory displacements. Isothermal scans for glassy samples were performed in a dry nitrogen atmosphere. The deformation control was set at 10 μm with a force limit of 0.9 N was applied an oscillating frequency of 10 Hz.

The same testing procedure was applied to the swollen wet polymer samples (without desiccation step) at 0.09 N force limitation; Hypotonic and isotonic equilibrated P-HEMA-nanosilica hydrogels with 5 to 25% by volume filler contents were isothermally tested at 37 °C in a 100% relative humidity nitrogen atmosphere at the frequency of 10 Hz.

### 2.3. Sorption and Swelling Tests

In total, 25 dry specimens of 5% loading nano-filled PHEMA of 2.5 mm thickness and 20 × 20 mm side length, prepared according to the conditions described in the previous section, were equilibrated in aqueous physiological isotonic and hypotonic saline solutions (0.15 and 0.05 M of NaCl). These two solutions were chosen to simulate two possible different conditions of physiological extracellular fluids and their influences on hydrogels sorption behaviours. [23,24].

The choice of using the 5% composition in the swelling tests arises from consideration about mechanical properties and biocompatibility considerations. In previous works we described in vitro tests with pHEMA with different composition of nanosilica [23]. Experimental evidence indicated that the hybrid systems developed significantly improved cell adhesion not only for murine fibroblasts but also for primary cultures of human osteoblast. The influence of the hybrid material nanosilica contents on cytocompatibility and cell adhesion indicated that optimal conditions were observed for a composition of 10% by weight (about 5% by volume according to the polymer and filler densities). It was then decided in the planning the swelling test to choose a material presenting a good compromise between mechanical and biocompatibility properties that could be reasonably found around this composition. In addition, this composition has also been tested in in vivo experiments on minipigs. Titanium implants modified with inserts in our hybrid swellable material with this composition have shown good bioactivity as bone scaffolding improving implant primary stability and osteointegration [19].

The plates, after immersion in the physiological solutions simulating the extracellular fluids, were monitored for sorption (by weight measurements) and swelling (by optical microscope measurements) kinetics. Equilibrium uptakes values in isotonic and hypotonic solution were statistically evaluated at equilibrium for all nanocomposite compositions.

Namely, samples’ weight gains due to the solution sorption in the initially dry samples were determined up to equilibrium by gravimetric measurements using a 0.1 mg resolution Mettler Toledo balance (Milan, Italy) and evaluated as a percentage increase in the initial dry sample weight.

The advancing swelling fronts associated with the anomalous sorption modes [13,14,15,16,17,18] and dimensional samples changes of the plate sample (thickness and side length increases along the Z-axis and the two orthogonal axes X and Y) were monitored as a function of the immersion time using an optical Leitz stereo-scan microscope. The sorption and swelling experiments were performed at 37 °C (thermostatic water bath) until constant weight equilibrium up-takes and dimensional changes were monitored (i.e., 50 hrs). Three samples were extracted from the equilibrating solution for each measurement at different times, rapidly weighed and measured and then returned to the conditioning solutions.

## 3. Results

In a previous work [5] we reported that glassy hydrophilic PHEMA-nanosilica hybrid composites undergo a complex sorption process when immersed in water, see Figure 1.

The diffusion of small penetrant molecules having a high chemical affinity with the host glassy polymer, in fact, is characterized by an anomalous sorption behavior [15]. The highly hydrophilic glassy PHEMA-nanosilica hybrids used in our study are, in fact, been described to strongly swell when immersed in physiologic simulating aqueous media [4,5].

The physical phenomena occurring in the initially glassy p-HEMA polymer once exposed to the aqueous medium have been described in reference [5] and described in Figure 1: the initially glassy hydrophilic sample is progressively externally swollen by the water sorption finally reaching an equilibrium in a fully swollen state (see right lower part of Figure 1). Initially, (see lower left part of Figure 1) a clear front separating an unpenetrated glassy core and a swollen outer shell is observed to progressively penetrating the sample glassy core.

The swelling process is accompanied by a significant increase in the sample water content leading towards final rubber hydrogels whose mechanical characteristics can be differently influenced by the composition of the equilibrating aqueous medium and by the presence of fillers in the hydrophilic matrix. A mechanical characterization on our hybrid nanocomposites at different nanofiller loading in their dry and hydrogel forms after equilibration in Isotonic and Hypotonic physiological solutions has been then carried out in order to understand the level and intensities of the solvent-induced physical changes.

The complex shear moduli measured in Dynamic Mechanical Analysis (DMA) tests run at 37 °C and 10 Hz on the glassy and hydrogels samples as a function of their nanosilica loadings are reported in the semi-logarithmic diagram of Figure 2.

As resumed in Table 1 where statistically evaluated mean values and standard deviations are reported, the measured values of the shear moduli range from 0.69 ± 0.04 to 8.0 ± 0.36 GPa for the dry glassy hybrid nanocomposites, and from 19.3 ± 2.1 to 112.3 ± 9.6 MPa and from 11.2 ± 1.7 to 75.5 ± 2.3 MPa for the hybrid hydrogels equilibrated in Iso-tonic and Hypo-tonic physiological solutions, respectively.

As indicated in previous publications [5,22], a not usual for particle reinforced composite linear increase in complex shear modulus of dry P-HEMA-nanosilica composites at increasing silica contents is observed. The shear modulus of the nanocomposite with 25% volume of nano-silica loading reaches a value (9.0 GPa) that is 10 times higher than the value of the dry unfilled P-HEMA (0.9 GPa). These significant increases in the shear moduli compared to the values expected applying the ordinary Halpin-Tsai equations approach [25] have been attributed to the hybrid nature of the resulting nanocomposites that is stiffened at segmental level by the entangled polymer structure interpenetrating the nano-silica inorganic network [26].

Characteristic ranges of variation of the shear moduli of cortical bone (2.5 ± 0.8 GPa) [27] and for articular cartilages (24 ± 3 MPa) [28,29] have been also reported on Figure 2 for comparison with the mechanical behavior of our materials.

It can be inferred that for a content of about 5% by volume of nanosilica, the hybrid nanocomposites in their glassy and hydrogel states show elastic properties comparable with those of cortical bone and human articular cartilage, respectively. This composition has been chosen as the optimal hybrid formulation for bone tissue engineering and used in the prosecution of the research described in this paper.

Bone tissue engineering can positively benefit the use of tuneable biomimetic scaffolding biomaterials that can be “designed” to act as biocompatible and biomechanically active hybrid interfaces [19,20]. The proposed biomechanical design approach to evaluate the most opportune material composition, however, should be integrated with an appropriate study on its response to physiological liquid variations. From Figure 2, in fact, it can be also inferred that these structural hydrogel systems are very sensitive to the composition of the physiological solutions. The shear moduli are, in fact, significantly influenced by these variations.

Further investigations on the kinetics and equilibria of the physical phenomena occurring on the hybrid nanocomposite have been performed on the chosen optimal composition of 5% by volume nanosilica-loaded P-HEMA.

The kinetic raw data for physiologic fluids simulating NaCl aqueous solution measured as sorption uptakes and residual unswollen glassy core for are reported in Figure 3 as a function of square root of time.

The sorption kinetics in the glassy samples from both equilibrating Hypo- and Iso-tonic solutions show a complex behavior characterized by a discontinuity at about 500 min (21 in the t^1/2^ axis of Figure 3).

The samples during the equilibration in the physiological solutions showed the formation of a clear swelling front separating the unaffected glassy core and an outer swollen layer (see schematization in Figure 3). By a parallel investigation carried out by measuring the kinetic of the advancement of the swelling front considering the thickness of the residual core during the equilibration test (reported on the left axis in the same Figure 3), it is evident that the discontinuity occurs when the residual glassy core disappears. In the presence of the glassy rigid core, in fact, free swelling is principally occurring along the z axis (see sample schematization in Figure 3). Once the constrain of the rigid glassy core in the X and Y directions disappears, the solvent-penetrated layers can relax and expand in those directions leading to additional sorption.

It is, therefore, evident that several physical phenomena are simultaneously occurring during the test. Namely, penetrant molecules diffusion, polymer swelling, time-dependent boundary conditions, and an abrupt change in the transport mechanism due to the polymer relaxation at the glassy core interface should be considered [12,13,14,15,16,17,18]. A brief description of the possible limiting transport processes is given in order to better understand the observed complex sorption behaviors.

Generally, penetrant diffusion in polymers above their glass transition temperature or in glassy polymers with a low chemical affinity with the diffusing species could be described by ordinary Fick’s law [12]. This first limiting transport process is characterized by a linear relationship between the initial weight gain and the square root of time, and it is characterized by a smooth and continuous concentration profile through the sheet thickness.

Conversely, a second limiting transport process, named Case II [16], occurring in thin glassy polymers films in contact with high chemical affinity diffusing molecules, is characterized by a step discontinuity of the penetrant concentration profile, that, other than being exclusively associated with Fickian diffusion, is principally driven by polymer relaxation. For this limiting Case II transport process, the weight gain is linear in time until equilibrium is reached. A discrete discontinuity, which separates the uniform penetrant concentration outer swollen shell and the glassy central core [13,14,15,16,17,18], is observed and moves through the samples at a constant rate.

In very general terms it can be stated that the initial sorption uptakes can be expressed as functions of time elevated to the exponent **n** (t^n^) with **n** = 0.5 (square root of time) for Fickian diffusion and **n** = 1 (linear in time) for Case II limiting sorption mode.

However, the penetrant molecules’ diffusive resistance generated in the outer swollen shell [24] can reduce the swelling front advancement rate [24,30,31,32,33]. Thicker samples, such as those used in our experiments, in fact, can show a still more complex sorption behavior since the diffusive resistance in the outer swollen shell reduces the concentration of solvent at the glassy core interface, lowering the sorption process. In fact, the diffusive resistance generates a smooth penetrant concentration profile in the swollen outer shell with its minimum at the swollen-glassy interface producing a lower plasticization of the polymer and, consequently, a slower penetration rate [27].

The water sorption expressed as an initial dry weight percentage increase as function of the square root of time in Figure 2, clearly shows an anomalous behavior where it is difficult to differentiate between the two previously described limiting sorption modes (Fickian and Case II). For this reason, swelling advancing fronts have been plotted as a function of the time in Figure 4.

The initially swelling kinetics (dotted lines in Figure 3) for isotonic and hypotonic aqueous solutions are, respectively, 11.0 mm/min and 9.6 mm/min.

The penetration rates progressively slow down as the swelling front advances in the unpenetrated glassy core indicating that a diffusion resistance is developing in the outer swollen thickness. It can be, then, inferred that limiting Case II sorption occurs only in the early first stages of the solutions equilibration process.

If a water uptake vs time diagram using log-log scales is used, the slope of the curve is the exponent *n* of the previously cited function of time **t^n^**. In Figure 5, where the sorption uptakes have been plotted in a log-log diagram, two linear portions characterized by slopes **n** = 1 and **n** = 0.5 are present.

The first portion of both curves with slope **n** = 1 represents the condition of constant rate swelling front advancement (limiting Case II sorption). In these conditions of constant rate sorption it can be stated that the process is only governed by the osmotic pressures generated at the swelling front where the penetrant concentration is constantly at its equilibrium value [30,31,32].

After 8 to 10 min, corresponding to a penetrated thickness of the 0.09 to 0.10 mm, the swelling process starts to become diffusion-controlled (slope **n** = 0.5). In physical terms, this means that the penetrant molecules concentration at the swelling front is lowered by the diffusive resistance encountered by the penetrant molecules in the progressively thicker external swollen layer [30,31,32].

The lower concentration of the swelling medium reduces the corresponding osmotic pressures at the interface, progressively reducing the rate of advancement of the swelling process until glassy core disappearance (after 500 min as indicated by an arrow in Figure 3). At longer times up to final equilibrium (about 2500 min), the sorption uptakes observed in Figure 3 are due to the residual diffusion in the not fully saturated but swollen polymer.

Measurements of the final equilibrium dimensional changes in both equilibrating media resulted in linear expansions along the X, Y and Z axes of the order of 15%, 15% and 17%, respectively.

## 4. Discussion

The sorption modes and physical phenomena occurring during the equilibration in physiological solutions of the initially glassy nanocomposite hybrids are to be attributed to the high affinity between the proposed biomaterials and the physiological liquid simulating solutions. Changes of the polymer water penetrant molecules activities and osmotic pressures in water solution sorption tests at increasing solute molarity have been investigated using the Flory Huggins theory approach [31].

The Flory-Huggins interaction parameter **χ** measures the interactions of the polymeric chain segments with the penetrant molecules as well as the polymer-polymer interaction [31]. The free energy of mixing of polymer chains with penetrant molecules was described by Flory-Huggins/ Rehner theories [33,34] considering the two contributions of the enthalpy of mixing, ΔH_mix_, and the entropy of mixing, ΔS_mix_. The entropy of mixing is calculated by the volume fractions of solvent and polymer, while the enthalpy of mixing is determined by a dimensionless interaction parameter **χ**. The presence of a second component in a polymer solid solution lowers the chemical potential of the host molecule from its value in the pure condition. The evaluation of the chemical potential changes in differently water moisture-saturated polymers is needed to calculate the osmotic pressure rise due to the penetrant molecule sorption.

A theoretical expression for the reduction in the chemical potential of the penetrant molecules is derived from the free energy of mixing since the chemical potential of a penetrant molecule (solvent) in non-ideal solutions relative to that in its pure state relates to its activity through the free energy of mixing [17,18,33,34],
μ_1_ − μ_1_° = RT ln a_1_ = RT [ln (1 − ϕ_2_) + (1 − 1/x) ϕ_2_ + χ_1_ϕ_2_^2^](1)
where μ_1_° is the pure penetrant molecule chemical potential, μ_1_ is penetrant molecule chemical potential in solid solution with polymer, R is the gas constant and T the absolute temperature. Moreover, a_1_ is the activity of the dissolving molecules (which in our experiments was the water activity in the NaCl saline solutions), ϕ_2_ is the volume fraction of the polymer in the polymer-water equilibrium solution at different penetrant water activities.

The interaction parameter χ_1_ accounts for the enthalpy changes occurring when polymer-polymer and penetrant-penetrant interactions are replaced by polymer-penetrant interactions.

Finally, x is the average degree of polymerization, x = *V*_2_/*V*_1_ with *V*_1_ and *V*_2_ the molar volumes of penetrant and polymer, respectively. This term, which is very big for polymers/water systems since representing how many solvent molecules are needed to form the polymer macro-molecule or crosslinked network, allows negligible 1/x values in Equation (1) that can be then written as:ln a_1_= ln (1 − ϕ_2_) + ϕ_2_ + χ_1_ϕ_2_^2^(2)

This equation correlates solution water activity in the sorption tests and the experimentally measured water uptakes.

Using the experimental equilibrium sorption uptakes, in fact, we may evaluate the interaction parameter, which should theoretically depend on the penetrant molecule concentration, at different water uptakes in the amorphous PHEMA/nano-Silica Hybrid material. The calculated interaction parameters χ_1_ at two different equilibrium water uptakes at 37 °C are reported in Table 1.

By knowing the interaction parameters χ_1_ we may evaluate the osmotic expansion stresses generated by the presence of the water molecules sorption at different equilibrium moisture contents using this derivation:σ = π = −(μ_1_ − μ_1_°)/V_1_^0^(3)
where V_1_^0^ is the water penetrant molecules molar volume (i.e., 18.07 mL mol^−1^·MPa^−1^).

Referring to Equation (2) considering the water volume fraction, and assuming the contribution of the 1/x (the ratio between the molar volumes of the hosting polymer and the water penetrant) negligible, we will obtain from Equation (4) the osmotic swelling pressures for Hypo and Isotonic solutions reported in Table 2.
π = −RT/V_1_^0^ [ln (1 − ϕ_2_) + ϕ_2_ + χ_1_ϕ_2_^2^] (4)

The calculated osmotic swelling pressures/stress range from −2.5 GPa at the lower equilibrium water contents (volume fraction 0,41 when saturated in iso-tonic NaCl water solution) to a maximum of −3.6 GPa at higher equilibrium water contents (final water uptake of 44.2% corresponding to 0.35 volume fraction when saturated in hypo-tonic NaCl water solution).

In Flory’s theory [33], the isotropic stress generated by the absorbed molecules equates to the total osmotic pressure to reach a static equilibrium state in the polymer. Namely, the penetrant swelling stress (σ) in the polymer exerts a compressive pressure (π) on the penetrant molecules. “…*A close analogy exists between swelling equilibrium and osmotic equilibrium. The elastic reaction of the network structure may be interpreted as a pressure acting on the solution, or swollen gel*…” [33].

The experimental values of the shear modulus in our DMA characterizations at 5% by volume PHEMA-nanosilica hybrids material (which is the composition used in our sorption-swelling tests) is 2.0 GPa (see Figure 2). However, in order to compare osmotic pressure generated by the sorbed molecules in the hosting polymeric system, the material characteristic bulk modulus (hydrostatic compression rigidity) should be calculated. The relationships between Elastic, Shear and Bulk moduli depends on the material Poisson’s ratio [35,36]. For most engineering plastics the Poisson’s ratio in the range 0.35 < ν < 0.45 and then, for quasi-isotropic crosslinked polymers and other glassy isotropic polymers, the elastic modulus equals the bulk modulus E ≈ B while the ratio between the bulk and shear moduli is E ≈ B ≈ 8/3 G.

In the case of our PHEMA-nanosilica hybrids, the calculated bulk modulus B (elastic behavior under hydrostatic pressure) is 5.3 GPa that is of the same magnitude of the values of the osmotic pressures calculated from the sorption data for isotonic (−2.4 GPa) and hypo-tonic (−3.6 GPa) solutions.

The difference Δ between the equilibrium internal osmotic swelling pressures generated at equilibrium by the absorbing species (essentially water molecules) and the material characteristic bulk modulus (hydrostatic compression rigidity) is reported in the last column of Table 1, Δ_Hypo_ = −1.7 and Δ_Iso_ = −2.8 for hypo-tonic and iso-tonic aqueous solutions, respectively.

It can be inferred from Table 1 and sorption/swelling kinetics (evaluated as mm/min from Figure 4) that the lower the difference Δ = B–π between material bulk modulus and internal osmotic pressure induced by the absorbing species, the higher and faster are the sorption uptakes, water diffusivity and swelling rates.

## 5. Conclusions

Our study was conducted on dense non-porous materials. The aim of the research was, in fact, to investigate the potentially tunable characteristics of this class of hydrofilic hybrid materials. Specific scaffold applications with targeted porosity can be obtained by using porogens such as cyclohexanol, dodecan-1-ol or saccharose [37,38]. Moreover, the impact of the of the hybrid material nanosilica contents on its biocompatibility and cytotoxicity has been investigated in our previous works [19,23] where optimal conditions were observed for composition between 5 to 10 %volume. It can be stated that a good compromise between mechanical and biocompatibility properties could be reasonably found in this range of compositions.

Hybrid ceramo-polymeric nanocomposites based on Hydroxyl-Ethyl-Methacrylate polymer (P-HEMA) filled with 7–15 nm fumed nanosilica particles (5% by volume) is a good candidate for a biomimetic material in bone tissue engineering. This material in the presence of simulating hypotonic and isotonic physiological liquid environments turns from rigid glass to rubbery hydrogel swelling up to 50% of its initial volume (14% linearly) and absorbing up to 40–45% by weight (depending on the composition of the external physiological medium).

The dynamic mechanical analysis performed on the exsiccated and on physiological-like liquid-swollen samples has shown that the elastic behavior of this hybrid material in its dry glassy state at the chosen filler composition is comparable with that of bone while, when hydrated to its rubbery hydrogel state, it possesses elastic properties comparable to those of the cartilage.

The use of mechanically tuned biomimetic hybrid hydrogels as scaffolding materials is expected to further improve prosthesis adaptation mechanisms since introducing active interfaces that adapt implant biomechanics for better reproducing cartilage and ligaments functions [6,7,19,20,21,22]. Adaptive properties of bone could benefit the use of biomimetic (biomechanically compatible and bioactive) scaffold biomaterials coupled with new designed prostheses.

A biomimetic/physiological approach can be then pursued in designing custom formulations of these new tunable ceramo-polymeric hybrid structural hydrogels that better adapt to the patient bone characteristics acting as bioactive and biomechanical stimulation and potential improved bone scaffold mineralization and ossification.

Sodium ions concentration changes can induce modification of the physical and mechanical properties of hydrogels used for structural or scaffolding systems.

Our hybrid structural hydrogels are designed to act as biocompatible and mechanically bioactive scaffolds for implant with the scope of favoring adaptive directionally organized bone growth. In order to achieve this result, both a proper scaffolding biocompatible and biomechanically active material has to be designed.

The biomimetic characteristics of our hybrid initially glassy materials have been investigated both for mechanical and swelling properties in order to better understand all the physical and morphological modifications occurring when exposed to mutating physiological like conditions.

Although model physiological-like solutions have been tested, our results have shown that the swelling process depends on water activities and the induced osmotic pressures generated in the hydrogels. Water wets biological surfaces and hydrates organic and synthetic polymeric compounds under the effect of a single driving force, namely, the gradient of its chemical potential.

This phenomenon has been described and related to water activities with Equations (1)–(4) in our theoretical approach that considered water activity changes in physiological-like solutions containing different concentrations of Sodium Chloride.

For physiological fluids containing proteins other than solvated salts, the presence of co-solutes changes water activity and therefore their chemical potentials and resulting osmotic pressures generated in the hydrated systems. Water activity of the external equilibrating solution containing proteins to be considered in our model can be evaluated through experimental methods measuring water fugacity and its partial pressure [38,39,40].

Moreover, it has to be considered that due to the significant differences in diffusivities between water molecules and protein macromolecules [41,42], which in collagens have been related to their hydrodynamic radii and reported to be of the order of 10^−8^ cm^2^/s for proteins and 10^−6^ cm^2^/s for water ionic solutions [43,44,45,46], water and ionic species are only expected to diffuse in our hydrating and swollen hybrid nanocomposites.

In the case of use as scaffolding material for bone tissue engineering, physiological bone material behavior to be mimicked by the bioactive scaffolding material are related to the chemical and mechanical properties that can be achieved using hybrid structural hydrogels.

Mechanical properties in the dry and swollen states as a function of the changing physiological fluid water contents, as well as bioactivity and swelling behavior sensitivity to external physiological modifications could play a relevant role in designing smart and biomimetic hydrogels. Hybrid ceramo-polymeric insert swelling could be used to stabilize implants in the bone and to create a biomechanically active interface that favors bone growth [19].

Stresses on the bone can be modulated by proper choice of scaffold swelling thickness and structural properties for healthy bone growth. In vivo tests performed using these new modified oral implants confirmed the improved capability of such hybrid hydrogels in promoting early osseointegration [19,22,29].

Moreover, aspects related to achieving material-controlled biodegradation could be further investigated for these materials. The present study has been conducted on crosslinked poly (2-hydroxyethyl methacrylate) (pHEMA). Previous investigations of in vitro biodegradation with J774.2 cells of cross-linked and linear pHEMA are reported in the literature [44]. Macrophages have been described in this study to be able to erode the surface of linear pHEMA but unable to erode the surface of the cross-linked polymer. Cells appeared wrapped by the linear pHEMA surface, while those cultured on the cross-linked polymer were only lying at the surface without evident interactions. This behavior has been confirmed for our crosslinked pHEMA hybrid materials tested for 2–4 months in in vivo experiments that did not show under microscopy any evident degradation effect [19].

Further studies on linear pHEMA and different formulations of biodegradable crosslinked nano-composites based on poly (2-hydroxyethylmethacrilate) and Polycaprolactone (pHEMA-PCl) copolymers are in course to evaluate the potential tunable biodegradability of such systems.

## Figures and Tables

**Figure 1 bioengineering-08-00045-f001:**
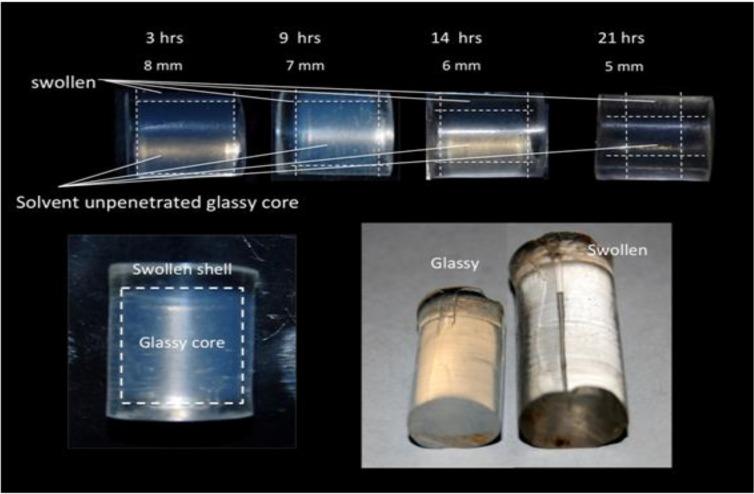
Swelling behavior of a hybrid hydrophilic nanocomposite in water [5]: Volumetric swelling from glassy to rubber states (Lower right), clear front between the internal unpenetrated glassy core and the water-swollen outer shell (Lower left), the front progressively advances at increasingly reducing the glassy core thickness (Upper part of the figure).

**Figure 2 bioengineering-08-00045-f002:**
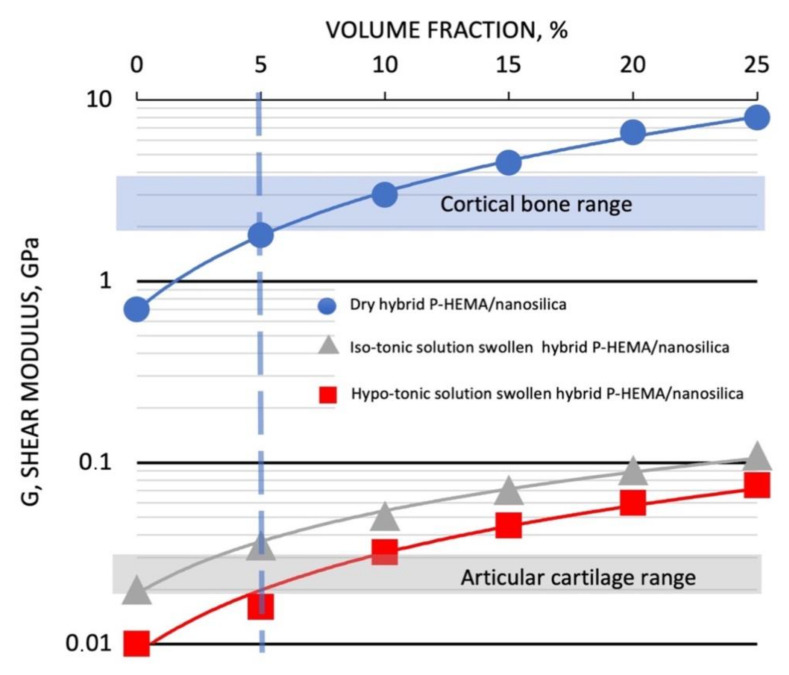
Dynamic Mechanical Analysis (DMA) Complex Shear Moduli of Dry (Blue circle), Hypotonic equilibrated hydrogels (Grey triangles) and Isotonic equilibrated hydrogels (Red squares) of P-HEMA-Nanosilica hybrid nanocomposites at increasing nanosilica loadings. Cortical bone and articular cartilage ranges of variation of shear modulus are also reported on the figure.

**Figure 3 bioengineering-08-00045-f003:**
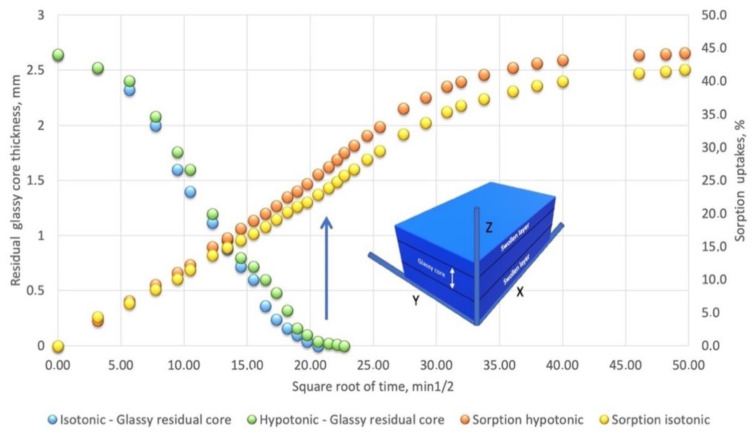
Swelling kinetics measured (left axis) as residual glassy core thickness and anomalous sorption behavior (right axis) in initially glassy 5% by volume nanosilica-loaded PHEMA nanocomposite equilibrated in hypotonic and isotonic physiological solutions.

**Figure 4 bioengineering-08-00045-f004:**
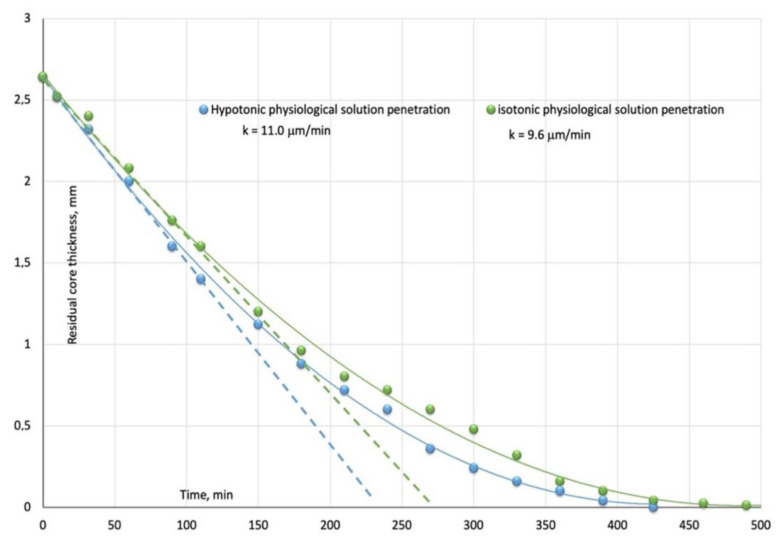
Swelling kinetics measured as residual glassy core thickness at different immersion times for initially glassy 5% by volume nanosilica-loaded PHEMA hybrid nanocomposite equilibrated in hypotonic (blue square) and isotonic (orange circles) physiological solutions. Dotted lines represent the initial penetration rates.

**Figure 5 bioengineering-08-00045-f005:**
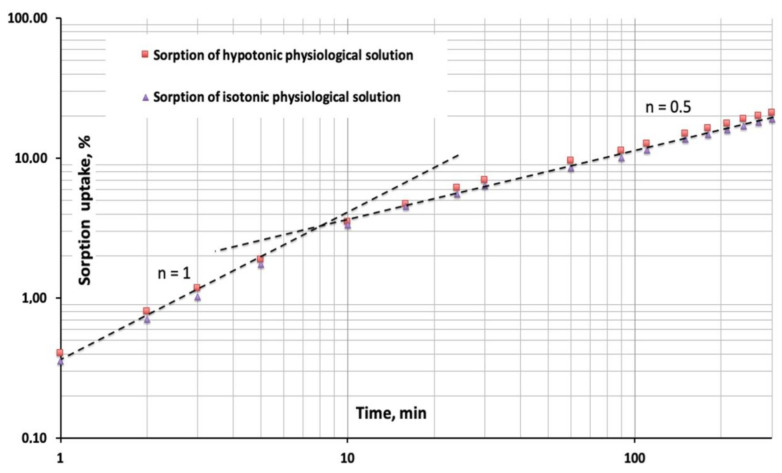
Early stages of the swelling process plotted in a log-log scale diagram. The slope **n** = 1 represents the limiting Case II sorption mode while the slope **n** = 0.5 represents a diffusion-controlled process (swelling front advancement) occurring during the equilibration in physiological solutions of the 5% by volume nanosilica P-HEMA hybrid material.

**Table 1 bioengineering-08-00045-t001:** Shear Moduli of dry and swollen in Iso- and Hypo-tonic physiological solutions pHEMA/nanosilica hybrid nanocomposites.

Composition % by Volume	Dry Shear Modulus GPa	Isotonic Wet Shear Modulus MPa	Hypotonic Wet Shear Modulus MPa
0	0.69 ± 0.04	19.3 ± 2.1	11.2 ± 1.7
5	1.73 ± 0.23	33.7 ± 2.1	16.4 ± 1.7
10	3.07 ± 0.35	51.0 ± 3.6	32.3 ± 1.0
15	4.43 ± 0.35	69.7 ± 4.0	43.3 ± 2.5
20	6.50 ± 0.46	90.7 ± 4.0	60.1 ± 1.3
25	8.00 ± 0.36	112.3 ± 9.6	75.5 ± 2.3

**Table 2 bioengineering-08-00045-t002:** Osmotic pressures and sorption properties of Hybrid Hydrogels ^1^.

Swelling NaCl Saline Solution	ϕ_1_ Equilibrium Volume Fraction	χ_1_ Flory-Huggins Interaction Parameter	π, GPa Osmotic Swelling Pressure	Δ GPa Bulk Modulus—Osmotic Pressure	k mm/min Swelling Rate (Case II Mode)
Hypotonic	0.35	0.91	−3.6	Δ_Hypo_ = −1.7	11.3
Isotonic	0.33	0.86	−2.4	Δ_Iso_ = −2.8	9.8

^1^ 5% by volume P-HEMA-nanosilica hybrid hydrophilic nanocomposite.

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
