# Peer review of "Biomechanically Tunable Nano-Silica/P-HEMA Structural Hydrogels for Bone Scaffolding"

_bioengineering, 2021, doi:10.3390/bioengineering8040045_

Round 1
Reviewer 1 Report
The manuscript is interesting; it concerns the preparation of hybrid hydrogels containing PHEMA and different amounts of nanosilica glassy. Likewise, the Authors focused on their swelling and dynamo-mechanical properties. However, in my opinion, this manuscript includes some inconsistences that require correction:
- Despite the fact that the introduction of the manuscript includes new scientific references, I still find it incomplete. As far as I am concerned, it should be enriched with novel information about hydrogels, which can be used in bone scaffolding.
- In the point 2. which discusses the preparation of hydrogels-based PHEMA and nanosilica glassy, I was not able to find any information about the amount of initiator (AIBN) and cross-linker used. Authors should describe this part in more detail.
- Fig. 3, 4 and 5 present only the average values of results. Therefore, the question which arises here is: what about statistical calculations? The standard deviations allow to show the amount of variation or dispersion of obtained results, therefore, I strongly recommend the Authors to enrich this part with the statistical analysis of their data.
- I understand the Author’s selection of sample containing 5% by volume nanosilica in PHEMA hydrogels on the basis of DMA analysis. It was presented clearly. However, I am not sure whether it is sufficient? Authors should have used some additional method to confirm this choice. It requires further explanation. On the other hand, the whole manuscript was based on the sorption and swelling kinetics, but only one sample was taken into consideration. Obviously, both the results and discussion are very interesting, but perhaps it would be worthwhile to exhibit some trend in this research for the entire series of samples.
Reviewer 2 Report
Aversa et al. investigated use of a biomimetic hybrid hydrogel made out of amorphous fumed silica nanoparticles embedded in poly- (hydroxy-ethyl-methacrylate) (pHEMA) enabling mechanical transition from a hard material in dry glassy state to a softer rubbery state upon hydration under physiologically relevant tonicities. The authors provide good insights into mechanical measurements and sorption kinetics which correlate with the fundamentals of polymer physics. Overall, the study is highly relevant to the field, especially to bone scaffolding. However, some concerns about the biological aspects of the hybrid hydrogel should be addressed.
- Authors should provide information about the pore size distribution in the hydrogel as a function of weight fraction of silica. This becomes highly relevant because the scaffold is expected to interface with cells and if weight fraction would have any impact on the intended application.
- Authors should explain how they expect the swelling behavior to change due to presence of proteins in the fluid that is expected to be surrounding the implanted scaffold. Would adsorption of protein have any impact?
- Authors should comment on the biodegradability of this hybrid material.
- Acronym OB (line 57) to be defined.
Author Response
"Please see the attachment"

Reviewer 3 Report
Aversa et al. have been investigated various hydrogel for their physical and mechanical properties. Their manuscript gave benefits of use of tuneable biomimetic scaffold biomaterials that can be “designed” to act as biocompatible and biomechanically active hybrid interfaces for bone tissue engineering.
This result will be very useful for tissue engineers to consider designing the material for tissue specific purposes through various physical properties. However, design is only limited for material part. Cellular interaction should be considered for tissue engineering purposes. There’s not much information of this material with cellular interaction in this manuscript. For the tissue engineering purposes, in vitro or in vivo assays is needed.
Round 2
Reviewer 1 Report
Thank you for your detailed response to my comments. I would like to recommend your article for publication in the Bioengineering.